# The Dose/Fractionation Debate in Limited-Stage Small Cell Lung Cancer

**DOI:** 10.3390/cancers16101908

**Published:** 2024-05-17

**Authors:** Kaixin Du, Xuehong Liao, Kazushi Kishi

**Affiliations:** 1Department of Radiation Oncology, Xiamen Humanity Hospital, Fujian Medical University, Xiamen 361004, China; dukaixin9022@gmail.com; 2Department of Pathology, School of Medicine, Sapporo Medical University, Sapporo 060-8556, Japan; lxh900921@gmail.com; 3Department of Radiation Oncology, National Disaster Medical Center, National Hospital Organization (NHO), Incorporated Administrative Agency, 3256 Midori-cho, Tachikawa-City 190-0014, Japan

**Keywords:** small cell lung cancer, limited-stage, dose, fractionation, radiation therapy

## Abstract

**Simple Summary:**

The selection of radiation dose/fractionation schemes for limited-stage small cell lung cancer (LS-SCLC) is highly controversial. It mainly involves the impact of total biologically effective dose, overall treatment time, and fractionation dose on survival and toxicity. This article comprehensively analyzes current relevant research and discusses the advantages and disadvantages of conventional-dose/fractionation radiotherapy (ConvTRT), hyperfractionated radiotherapy (HyperTRT), hypofractionated radiotherapy (HypoTRT), and stereotactic body radiotherapy (SBRT), in order to find the most suitable dose fractionation scheme and provide reference for clinical practice. Regardless of the radiotherapy modality, controlling the radiotherapy time to around 3 weeks is beneficial for improving survival and local control rates while reducing toxicity reactions. ConvTRT with a high total prescription dose is not recommended, because a prolonged radiotherapy time may reduce the tumor control probability of some rapidly proliferating tumor cells. In balancing acute reactions, late reactions, and treatment outcomes (local control or overall survival), the advantages of HyperTRT over HypoTRT still exist. For the radical radiotherapy regimen recommended by current guidelines (45 Gy/30 fractions), we suggest moderately increasing the fractionation dose and total dose, which may further improve prognosis.

**Abstract:**

To explore the most suitable dosage regimen for limited-stage small cell lung cancer (LS-SCLC) and provide references for clinical selection, strict inclusion criteria were applied, and studies were screened from Pubmed, Embase, and Web of Science. Subsequently, data on two-year overall survival rates and dosage regimens were collected, and scatter plots were constructed to provide a comprehensive perspective. The survival benefits of various dosage regimens were evaluated, and a linear quadratic equation was utilized to fit the relationship between the biologically effective dose (BED10) and the two-year overall survival rate. Among the five randomized controlled trials, the two-year overall survival rate of ConvTRT regimens with BED10 > 60 Gy (rough value) was only at or below the median of all ConvTRT regimens or all included study regimens, indicating that increasing the number and total dose of ConvTRT does not necessarily lead to better prognosis. In the exploration of HypoTRT regimens, there was a linear positive correlation between BED10 and the two-year overall survival rate (*p* < 0.0001), while the exploration of HyperTRT regimens was relatively limited, with the majority focused on the 45 Gy/30 F regimen. However, the current 45 Gy/30 F regimen is not sufficient to control LS-SCLC, resulting in a high local recurrence rate. High-dose ConvTRT regimens have long treatment durations and may induce tumor regrowth which may cause reduced efficacy. Under reasonable toxicity reactions, HyperTRT or HypoTRT with higher radiotherapy doses is recommended for treating LS-SCLC.

## 1. Introduction

Small cell lung cancer (SCLC) accounts for approximately 15% of all lung cancer cases, with limited-stage SCLC comprising about 30% of SCLC cases [1]. Thoracic radiotherapy (TRT) combined with chemotherapy is considered the standard treatment for inoperable limited-stage small cell lung cancer (LS-SCLC) [2]. The main TRT (thoracic radiotherapy) regimens include hypofractionated TRT (HypoTRT, administered once daily with a fraction size >2.2 Gy), conventional TRT (ConvTRT, administered once daily with a fraction size ranging from 1.8 to 2.2 Gy), hyperfractionated TRT (HyperTRT, administered low fractions sizes twice or more daily), and stereotactic body radiotherapy (SBRT). To date, there has been intensive debate over the selection of radiation dose/fractionation without undergoing concurrent chemoradiotherapy; no consensus seems to be provided. We supposed that this was due to the limited comparison of dose schemes in each article. It is challenging to thoroughly explore the merits and demerits of these three forms with different dose/fractionation patterns in radiotherapy. Therefore, we reviewed and analyzed the results of published randomized controlled trials (RCTs), real-world cohorts, and single-arm trials, integrating the survival outcomes of different TRT schemes into a unified perspective method. This employed the biologically effective dose (BED10) as the *x*-axis and the two-year survival rate as the *y*-axis, delving into the acute and chronic toxicity reactions in articles, and then discussing the pros and cons of those different radiation dose/fractionation schemes to provide guidance for the selection of radiation therapy for LS-SCLC.

## 2. Materials and Methods

To provide a more intuitive assessment of the efficacy of different radiation fractionation schedules, this study collected dose fractionation schemes and survival data from various studies on LS-SCLC published up to 6 September 2023, from the Pubmed, Embase, and Web of Science databases. Double-blind standardized screening was conducted according to the ASTRO clinical practice guidelines [2] (Kaixin Du and Xuehong Liao). Investigator Kazushi Kishi subsequently reviewed the extracted data from all included studies and discussed discrepancies with other investigators to achieve consensus. We excluded abstracts without full texts, articles where most patients (more than 50%) did not undergo concurrent chemoradiotherapy, articles that did not apply concurrent platinum-based chemotherapy, and articles in which radiotherapy interventions were not administered within four chemotherapy cycles. Patients with advanced-stage diseases undergoing palliative radiotherapy were excluded from this study. Studies with fewer than 25 patients were also excluded. All studies implemented prophylactic cranial irradiation (PCI) for patients with good treatment response after chemoradiotherapy. The PCI regimens consisted of either 25 Gy/10 F or 30 Gy/10 F. The calculation formula for the biologically effective dose (BED) is as follows [3]:BED = (nd){1 + [d/(α/β)]} − (0.693t/αTpot)where n is the total number of fractions; d is the dose per fraction (Gy); α/β is 10; α is 0.3 Gy; t is the total number of days in which RT is delivered; and Tpot is the potential doubling time (5.6 days).

To gain a more accurate understanding of the overall effectiveness of different dose regimens in current research, we constructed scatter plots with BED10 on the *X*-axis and 2-year overall survival rate on the *Y*-axis, utilizing a linear quadratic equation for fitting [4,5], and we computed the goodness of fit (R^2^) and *p*-values. The BED10 data were derived from treatment regimens reported in studies after uniform calculation. In cases where multiple treatment regimens were used in a study, priority was given to the primary treatment regimen. The 2-year overall survival rates were obtained from either the provided study data or survival curves generated from this study. Some outcome values were extracted from survival curves using Engauge Digitizer (version 11.3).

## 3. Statistical Analysis

Ordinary least squares (OLS) was applied to establish a multivariate polynomial regression model. The quadratic linear regression equation was applied to simulate the correlation between 1-year LC and 1- and 2-year OS, and the fitting figures were drawn using the ggrepel package (version 0.9.1) and tidyverse package (version 1.3.2).

## 4. ConvTRT (Conventional Dose/Fractionation Radiotherapy)

Early concerns regarding fractionated doses around 2 Gy causing severe side-effects stemmed from limitations in two-dimensional radiation technology. Hence, conventional fractionation doses typically remained within 1.8–2.2 Gy. Additionally, different tissues respond to radiation in distinct ways, categorized into early- and late-reacting tissues based on their response time. Early-reacting tissues, such as the skin and mucosa, respond earlier due to their rapid cell proliferation and turnover rates. Using smaller fractionation doses (approximately 2 Gy) and appropriate radiation intervals gives these tissues sufficient time for regeneration, thus mitigating acute toxic side-effects. In contrast, late-reacting tissues exhibit a slower response to radiation doses, with damaged cells often unable to self-renew for several months or even over a year, subsequently undergoing compensatory proliferation [6].

The CALGB 30610 (Alliance)/RTOG 0538 phase III randomized controlled trial conducted by Bogart et al. [7] in 2023 included all patients randomly assigned to receive either 45 Gy twice daily (HyperTRT, n = 313, BED10 = 43.91 Gy) or 70 Gy once daily (ConvTRT, n = 325, BED10 = 64.61 Gy). The median follow-up at 4.7 years showed no improvement in overall survival (OS) in the highly biologically effective dose group (70 Gy once daily) (hazard ratio 0.94; 95% CI 0.76 to 1.17; *p* = 0.594). Survival curves indicated a 2-year survival rate of approximately 56% and a 5-year survival rate of 32% (95% CI, 27–39) in the once-daily group, compared to a 2-year survival rate of 59% and a 5-year survival rate of 29% (95% CI, 23–35) in the twice-daily group. Results from a single-arm trial with the same ConvTRT treatment regimen (70 Gy once daily) published by the same team 20 years ago also demonstrated similar survival outcomes, with a 2-year overall survival rate of 48% (95% CI, 37–63) and a median follow-up time of 24.7 months.

In the CALGB 30610 (Alliance)/RTOG 0538 phase III randomized controlled trial [7], the frequency of severe adverse events (including esophageal and pulmonary toxicity) was similar between the two arms, with a higher incidence of grade 3 adverse events in the 45 Gy group compared to the 70 Gy group (29.7% vs. 23.7%, *p* = 0.0855), while grade 5 severe adverse events were higher in the 70 Gy group compared to the 45 Gy group (11 cases (3.4%) vs. 4 cases (1.3%), *p* = 0.0792). Furthermore, the 70 Gy group exhibited more common occurrences of leukopenia (58.8% vs. 50.2%, *p* = 0.0343), anemia (26.2% vs. 20.3%, *p* = 0.0882), and lymphopenia (16.3% vs. 9.5%, *p* = 0.0135). Overall, the 70 Gy once-daily group experienced slightly more severe hematologic toxicity compared to the 45 Gy group.

Another study, known as the CONVERT trial, is an open-label, phase III, randomized, superiority trial [8], investigating the efficacy of different radiation treatment regimens, 45 Gy/30 F (HyperTRT, BED10 = 43.91 Gy) and 66 Gy/33 F (ConvTRT, BED10 = 60.64 Gy), on survival. A total of 547 patients were enrolled and randomly assigned to either the twice-daily group (274 patients) or once-daily group (273 patients). With a median follow-up of 45 months, the results showed a two-year overall survival rate of 56% (95% CI 50–62) in the twice-daily group and 51% (95% CI 45–57) in the once-daily group (hazard ratio for death in the once-daily group, 1.18 [95% CI 0.95–1.45]; *p* = 0.14). Among patients assessed for radiation toxicity, there was no difference in grade 3 esophagitis and radiation pneumonitis between the two groups. However, in the late-toxicity assessment at 3 months, the 66 Gy/33 F group had 17% of patients experiencing grade 1–2 esophagitis and 2% experiencing grade 3 late esophagitis, while the 45 Gy/30 F group had 12% of patients experiencing grade 1–2 late esophagitis (*p* = 0.06). Eleven patients died from treatment-related causes (3 in the twice-daily group, 8 in the once-daily group). A propensity scores matching analysis [9] was used to study the survival efficacy among different doses of ConvTRT treatment regimens. The results showed that, after propensity matching, compared to the ≤54 Gy (BED10) ConvTRT group, the >54 Gy (BED10) ConvTRT group demonstrated significant improvements in three-year progression-free survival and overall survival rates (42.7% and 56.2%, respectively, both *p* < 0.001 and 0.003). According to the survival curves [9], the two-year overall survival rate was 61% for the ≤54 Gy ConvTRT group, while it was 72% for the >54 Gy ConvTRT group (as shown in Figure 1). The most common dose fractions in this propensity-matched study were 54 Gy/27 Fx (n = 120, accounting for 53.3%, BED10 = 49.54 Gy) and 60 Gy/30 Fx (n = 70, accounting for 31.1%, BED10 = 55.5 Gy).

Through a scatter plot drawn with BED10 on the *X*-axis and 2-year overall survival rate on the *Y*-axis [3,7,8,9,10,11,12,13,14,15,16,17,18,19,20,21,22,23,24,25,26,27,28,29,30,31,32,33,34,35,36,37,38,39,40,41,42,43,44,45,46,47,48,49,50,51,52,53,54,55,56,57,58,59] (The details of all studys were shown in Appendix A), it can be observed that the 2-year overall survival rates of the five ConvTRT studies with BED10 exceeding 60 Gy (rough value) are not high (roughly in the middle of the overall 2-year survival rates). These five studies all come from randomized controlled trials, two of which are in phase 3 with three in phase 2. The speculated reason for the low overall survival rate may be due to the prolonged treatment time, leading to tumor regrowth offsetting the survival benefits of increased doses. This also explains why the high-dose ConvTRT regimens, with BED10 far higher than the classical HyperTRT regimen, failed to improve the overall survival of LS-SCLC in two randomized controlled trials, while the >54 Gy group in the propensity score matching analysis still achieved significant improvements in three-year progression-free survival and overall survival rates compared to the ≤54 Gy group. This is likely because the total treatment time of the ConvTRT regimens in the two randomized controlled trials reached 45 days or more, and for rapidly proliferating small cell carcinoma, the radiation damage effect of 2 Gy fractionation doses has been offset by the accelerated proliferation of cancer cells. In the propensity score matching analysis, the main dose fractionation scheme for the >54 Gy group was 60 Gy/30 Fx, with a treatment time of only 40 days. The excessively long treatment time not only makes it difficult for patients to adhere to the treatment, as seen in the Bogart trial, where over 20% of patients failed to complete the 70 Gy treatment [7], but also increases the risk of toxicity-related deaths [8].

## 5. HypoTRT (Hypofractionated Radiotherapy)

Currently, with the advancement of radiotherapy equipment, radiation oncologists are becoming increasingly daring to attempt larger fractionated radiotherapy regimens. However, due to concerns about treatment-related toxicities (mainly esophagitis) and logistical considerations of treating patients twice a day, only 25% of radiation oncologists surveyed recently endorsed twice-daily radiation therapy (RT) [60]. In the era of two-dimensional radiotherapy, HypoTRT has been used in Canada as a safe and effective treatment method for LS-SCLC (40 Gy/15 fractions), achieving similar treatment outcomes to the classical regimen of 45 Gy/30 F, twice daily, with a 5-year survival rate of 22%; however, similar to HyperTRT, the cumulative risk of local recurrence at 3 years exceeds 50% [52], attributed to low radiation doses. Several studies have compared the treatment efficacy of HypoTRT and HyperTRT under similar BED10 conditions, including retrospective analyses [42], overlapping weighted analyses [49], and randomized phase II trials [44], with results indicating no definitive conclusion regarding the superiority of either treatment. Yan et al. [49] used a propensity score matching for overlapping weighting to balance observed covariates between the two radiation regimen groups (45 Gy/30 fractions, twice daily or 40 Gy/15 fractions, once daily). The median follow-up time was 20.4 months. Multivariate regression models evaluated OS, LRR risk, and any ≥3-grade toxicity, all showing no significant differences. Bjørn H. Grønberg et al. [44], in a randomized phase II trial published in 2016, compared the same two-dose fractionation schedules, with the results indicating that more patients in the twice-daily group achieved complete remission (OD: 13%, BID: 33%), but overall response rates were similar. There were no differences in one-year progression-free survival (PFS) (OD: 45%, BID: 49%; *p* = 0.61) or median PFS (OD: 10.2 months, BID: 11.4 months; *p* = 0.93). For severe toxicity, there were no significant differences between the two radiation regimens, including ≥3-grade esophagitis and pneumonia. Also, a multicenter, phase II randomized trial published in 2021, conducted by Qiu et al. [3], showed that moderate hypofractionated once-daily HypoTRT (65 Gy/26 F, BED10 = 66.4 Gy) demonstrated improved PFS and similar toxicity compared to HyperTRT (45 Gy/30 F, BED10 = 43.91 Gy) in LS-SCLC. The median follow-up time was 24.3 months, with a median PFS of 13.4 months (95% CI, 10.8–16.0) in the twice-daily group and 17.2 months (95% CI, 11.8–22.6) in the once-daily group (*p* = 0.031), and two-year PFS rates of 28.4% (95% CI, 18.2–38.6) and 42.3% (95% CI, 31.1–53.5), respectively. The incidence of ≥3-grade acute lymphocytopenia in the high-dose once-daily group was significantly higher than twice-daily group (71.7% vs. 40.2%; *p* < 0.001). Other toxicities between the once-daily and twice-daily groups were similar, including the incidence of ≥3-grade esophagitis (17.4% vs. 15.3%). Possibly due to an insufficient follow-up time, the estimated median survival in the twice-daily group was 33.6 months, while in the once-daily group, it was 39.3 months (*p* = 0.137), with only a trend of difference in *p*-value.

Additionally, a scatter plot was constructed with BED10 on the *X*-axis and 2-year overall survival rate on the *Y*-axis [14,16,42,44,48,49,52,53,54,55,56,57,58,59], indicating a strong positive correlation between BED10 in the HypoTRT treatment group and 2-year overall survival rate (R^2^ = 0.93, *p* < 0.0001, as shown in Figure 2). This highlights BED10 as a robust indicator for predicting overall survival rates in LS-SCLC, attributed in part to the sufficient diversity in discussing HypoTRT dose regimens and the development of the calculation formula for biologically effective doses. Furthermore, the relationship between the biologically effective dose and late toxicity reactions also needs attention.

## 6. HyperTRT (Hyperfractionated Radiotherapy)

Studies have shown that, under the same total treatment time, late-reacting tissues are more sensitive to dose fractionation compared to early-reacting tissues [6]. In order to mitigate late toxicity reactions, Thames and others applied hyperfractionation to improved radiotherapy regimens following the recognition of the inherent differences in radiation response between early- and late-reacting tissues [6]. Since then, there has been a conscious effort to use hyperfractionation. SCLC cells are highly sensitive to radiation, as demonstrated by a soft agar cloning assay testing cell lines established from seven patients with SCLC, revealing exponential cell death even at low doses (extrapolated radiation curve D0 values of 1.0–3.3 Gy) [61], while low doses also reduce damage to surrounding normal tissues. A study in 1998 concluded that 45 Gy is the maximum tolerable dose for twice-daily thoracic radiotherapy in limited-stage SCLC [62]. A landmark study was the Intergroup 0096 trial, where fractionation and TRT intensity were tested, comparing 45 Gy TRT once daily over 5 weeks versus 45 Gy TRT twice daily over 3 weeks. The twice-daily regimen significantly improved overall survival, increasing the 5-year survival rate from 16% to 26%, leading to a paradigm shift in the treatment of LS-SCLC towards more routinely considering accelerated hyperfractionation.

Currently, most studies on HyperTRT focus on the classic regimen of 45 Gy/30 F, with 2-year overall survival rates ranging from 33% to 72.7%, with a median of 49% (Figure 1). A retrospective analysis by Tan et al. [29] in 2021 showed that 148 LS-SCLC patients attempted different dose regimens within the HyperTRT group, with 50 Gy/34 F twice daily, with an 8 h interval (BED10 = 48.01 Gy), while the remaining 74 underwent 56 Gy/28 F once daily (ConvTRT, BED10 = 51.53 Gy). The median overall survival (OS) after HyperTRT was 23.6 months compared to 20.2 months for the ConvTRT group, with two-year OS rates of 47% and 39%, respectively, showing a statistically significant difference (*p* = 0.032). The only article discussing dose exploration for HyperTRT is a randomized controlled phase II trial published by Bjørn Henning Grønberg in 2021 [47]. They analyzed dose regimens of 60 Gy/40 F twice daily (BED10 = 58.28 Gy) and 45 Gy/30 F twice daily (BED10 = 43.91 Gy). A total of 176 patients were enrolled, with 170 randomized to receive either 60 Gy (n = 89) or 45 Gy (n = 81). The primary analysis was a median follow-up of 49 months. After 2 years, 66 patients (74.5% [95% CI 63.8–82.9]) in the 60 Gy group were alive, the highest 2-year overall survival rate reported in all limited-stage SCLC studies. In contrast, 39 patients (48.1% [36.9–59.5]) in the 45 Gy group were alive (odds ratio 3.09 [95% CI 1.62–5.89]; *p* = 0.0005), similar to the median two-year overall survival rate for HyperTRT (49%). The increase in BED10 was only 14.37 Gy, resulting in a 26.1% increase in the two-year overall survival rate. The five-year survival rates were not reported, but they were estimated from the survival curves to be increased by 10%. The most common grade 3–4 adverse events were neutropenia (72 out of 89 patients [81%] in the 60 Gy group vs. 62 out of 77 patients [81%] in the 45 Gy group) and neutropenic infection (24 [27%] vs. 30 [39%]), possibly due to studies [63] showing that the granulocyte colony-stimulating factor increases the frequency and duration of life-threatening thrombocytopenia, leading to a significant increase in toxic deaths. Additionally, thrombocytopenia (21 [24%] vs. 19 [25%]), anemia (14 [16%] vs. 15 [20%]), and esophagitis (19 [21%] vs. 14 [18%]) were observed. There were 55 severe adverse events in 38 patients in the 60 Gy group [47], and 56 in 44 patients in the 45 Gy group. Three treatment-related deaths occurred in each group. The difference was not significant and was one of the lowest levels reported in limited-stage SCLC studies. There was no difference in progression-free survival between these two groups.

In Figure 1, the HypoTRT group reported by Qiu et al. [3] (65 Gy/26 Fx) achieved a good overall survival rate (74.2%, ranking second among the randomized controlled trials included in the analysis). However, analyzing these two studies, we found that the patients included by Qiu et al. were in better condition than those in Bjørn Henning Grønberg’s study. Qiu et al. [3] included only patients with an ECOG score of 0 or 1 and excluded those with an ECOG score of 2. So, the HyperTRT group (45 Gy/30 Fx) also had a good two-year overall survival rate.

## 7. Stereotactic Body Radiation Therapy (SBRT)

SBRT is a strategy that uses highly conformal techniques to deliver very high ablative radiation doses in 1 to 5 fractions to cancer targets, and it is not suitable for “supercentral” tumor patients [2], but it is suitable for elderly patients or those with limited physical fitness, histologically confirmed to be stage I to II, lymph node-negative, and located peripherally in SCLC patients. For early-stage SCLC patients with T1-2N0M0 who cannot tolerate or who refuse surgical treatment, stereotactic body radiation therapy (SBRT) can be considered [64,65,66], with dose fractionation schemes based on early NSCLC SBRT (BED ≥ 100 Gy). The utilization rate of SBRT increased from 0.4% in 2004 to 6.4% in 2013 among stage I patients [65]. A study from 24 different centers reported the largest-scale use of SBRT in stage I SCLC patients [67], evaluating 76 lesions in 74 patients, of whom only 59% received chemotherapy and >30% performed poorly (ECOG = 2–3). Nonetheless, the one- and three-year local control (LC) rates were 97.4% and 96.1%, respectively. Patients who received chemotherapy had significantly better efficacy, with a median DFS of 61.3 months compared to 9.0 months for those who did not receive chemotherapy (*p* = 0.02), and an OS of 31.4 months compared to 14.3 months (*p* = 0.02). Toxicity was rare, with 5.2% experiencing grade 2 or higher pneumonitis. Another study [64] included 29 limited-stage SCLC patients treated with all-synchronous etoposide + cisplatin chemotherapy. The median OS was 27 (95% CI 20.2–33.8) months. The median PFS was 12 (95% CI 4.2–19.8) months. No grade 4 adverse events were observed. Only five (13.8%, 5/29) patients experienced grade 3 adverse events. The estimated 2-year OS from the survival curve was 55.9%, and the estimated 5-year OS was 36%. A meta-analysis published in 2019 [68] included seven reports, finding that the ranges of 1-year, 2-year, and 3-year overall survival rates (OS) were 63% to 87%, 37% to 72%, and 35% to 72%, respectively. Distant metastasis was the most common failure mode, up to 38% to 53%. Toxic reactions were uncommon, with the most severe study reporting a 14% rate of grade 3 esophageal toxicity and a 7% rate of grade 3 hematologic toxicity.

Although SBRT can achieve excellent primary tumor control similar to NSCLC, the overall survival rate is still lower compared to inoperable NSCLC patients, with a three-year overall survival rate of 34% [69]. This may indicate the risk of SCLC to rapidly metastasize even when diagnosed early-stage and the limitations of radiation therapy in limited-stage SCLC, necessitating combination systemic treatment. Adjuvant chemotherapy is an indispensable treatment and significantly improves OS and DFS [70]. Although SBRT has been shown to be effective in the local control of stage I SCLC, further studies on larger cohorts, including the effects of combined chemotherapy, are needed [71]. However, although data are limited, and there are no randomized controlled trials of SBRT for treating SCLC, this approach is particularly useful for patients who are unsuitable for surgery [2].

## 8. Discussion

Traditional fractionated high-dose radiotherapy seems to offer no survival benefit for patients with LS-SCLC. Several large randomized controlled trials [7,8,19,26] suggest that the previously popular ConvTRT regimen may not confer survival benefits for patients when the BED10 exceeds 60 Gy (approximate value), possibly due to prolonged treatment time (6 to 8 weeks) leading to tumor repopulation, indicating that longer treatment time may not be conducive to better outcomes [72].

HyperTRT administers radiation twice a day, which can reduce each fraction dosage while maintaining the same overall treatment duration and aims to minimize the late complications without reducing treatment efficacy [5,6]. Arvidson et al. [73] also confirmed the benefit of twice-daily (BID) fractionation in achieving good two-year PFS than QD fractionation schemes. Although the HyperTRT regimen of irradiating twice a day may cause inconvenience to the patient, Grønberg et al. [47] claimed that no patient would give up treatment due to inconvenience. Qiu et al. [3] also confirmed that both daily HypoTRT and twice-daily HyperTRT have similar treatment compliance. However, they also suggested that daily HypoTRT may have more practical significance, including reducing healthcare resource utilization. Treatment-related toxicity is also a significant concern among oncologists. However, based on several large randomized controlled trials [3,7,8,11,37,44,74,75], compared to HypoTRT and ConvTRT, there is no significant difference observed in toxicity levels, including 3+ grade esophagitis and other 3+ grade toxicities.

The disadvantages of SBRT include high requirements for radiotherapy technology and equipment, as well as certain requirements for tumor location, limiting its indications. Additionally, in order to achieve ultra-high fractionated doses for tumor ablation, it is easy to induce large-cell transformation in SCLC [61], thereby increasing the risk of distant metastasis and treatment resistance. HypoTRT schemes may have advantages for combining immunotherapy in SCLC. Immunotherapy is a treatment method that stimulates the immune system to recognize and eliminate tumor cells. In the PACIFIC study, durvalumab (anti-PD-L1) was shown to improve PFS and OS outcomes in patients with locally advanced NSCLC after chemoradiotherapy [76]. Trials of consolidative immunotherapy after chemoradiotherapy are also underway in LS-SCLC. Hypofractionated radiotherapy can induce apoptosis or necrosis of tumor cells, releasing tumor antigens to stimulate the immune system’s attack on tumor cells, while immune checkpoint inhibitors can relieve tumor cells’ escape mechanism from the immune system, enhancing the killing ability of immune cells [77]. However, SCLC exhibits a strong immunosuppressive phenotype [78], and the effect of fractionated schemes on immunotherapy has not yet been verified by clinical trials.

The optimal fractionation regimen for radiotherapy should effectively control tumors with acceptable complications. TRT regimens of 45 Gy/30 F and 40 Gy/15 F have both been shown to have high rates of local recurrence, indicating that the radiation dose may need to be increased. A meta-analysis published in 2016 included 19 studies involving 2389 SCLC patients [79]. The results showed that higher BED prolonged mOS (R^2^ = 0.198, *p* < 0.001) and mPFS (R^2^ = 0.045, *p* < 0.001). A 10 Gy increment increased 1-year, 3-year, and 5-year OS by 6.3%, 5.1%, and 3.7%, respectively. Schild et al. [74] investigated the relationship between the five-year survival rate and biologically effective dose (BED10) reported in eight phase III randomized controlled trials from 1997 to 2004. A strong positive correlation was found between BED10 and the five-year survival rate in LS-SCLC (Pearson correlation coefficient 0.81). In our study, BED10 showed a highly linear positive correlation with two-year overall survival in the HypoTRT treatment group, even when fitted with a linear quadratic equation. These findings demonstrate the necessity of increasing the biologically effective during LS-SCLC treatment. As shown in the HypoTRT scatter plot (Figure 2), two-year survival was linearly correlated with BED10, and this correlation line did not seem to reach a plateau. This means that higher doses of HypoTRT may have better prognosis for LS-SCLC. However, due to the lack of exploration of treatment with high-dose HypoTRT or HyperTRT regimens, we cannot yet determine the appropriate dose range for LS-SCLC. But it is certain that the biologically effective dose recommended by the guidelines, such as 45 Gy/30 F, may be insufficient.

The short total treatment time is beneficial for overall survival. For LS-SCLC, a three-week treatment period may be reasonable. Studies have shown that the rate of tumor regrowth increases with time during radiation therapy or chemotherapy. For tumors with a very rapid growth rate, a short fractionation cycle is needed [6]. Shortening the time of local control during a week may increase overall survival by 7% to 10% [80]. Another study also found that tumors often begin to accelerate their growth after 3 weeks of treatment. For epithelial tumors, the probability of tumor control loss increases by about 1.0–1.5% for each day of treatment delay. The doubling time of surviving tumor cells during radiotherapy is estimated to be in the range of 4–8 days [72]. For small cell carcinoma, the doubling time is generally considered to be 5.6 days, indicating rapid tumor proliferation. Arvidson et al. [73] modeled the comprehensive effects of chemotherapy, incomplete repair of sublethal damage, and changes in the initiation time of rapid proliferation and effective tumor cell doubling time. They found that the optimal treatment duration for both fractionation schemes (HypoTRT and HyperTRT) was 3 weeks. The success of the classic scheme (45 Gy/30 F) may also prove this hypothesis.

In the past, limitations in radiation therapy technology resulted in smaller fraction doses, leading to extended overall treatment durations. While this favored normal tissue recovery and reduced acute toxicity reactions, it may have compromised the control of rapidly proliferating SCLC. Thanks to ongoing advancements in radiation therapy techniques such as stereotactic body radiation therapy (SBRT) and proton therapy, there is an enhanced preservation of normal tissues. This facilitates the increase in single radiation therapy doses, and the reduced toxicity may potentially confer overall survival benefits. Furthermore, it has the potential to modulate the tumor’s immune microenvironment, facilitating improved tumor recognition by the immune system through the release of tumor antigens induced by radiation therapy [81]. Arvidson et al. [73] used the dose and two-year progression-free survival rate reported in LS-SCLC studies to construct a dose–response model. The results showed that a more aggressive treatment with increased fractionation dose could significantly increase the current two-year PFS. The segmented dose increase model plotted on the basis of 2-year PFS and late complications normalized to the total dose showed that the BID model with a fractionation dose of 2 Gy and the QD model with a fractionation dose of 3.4 Gy reached the peak of 2-year PFS.

Regarding the best dose regimen, the current evidence has not yet reached a consensus, and dose regimens need to be tailored to the tumor’s location and the patient’s condition, necessitating individualized treatment. However, if we set aside the special circumstances of the tumor and the patient and focus only on the majority of patients with LS-SCLC, based on our research analysis, the dose regimen studied by Grønberg et al. [47], 60 Gy/40 F BID (BED10 = 58.28 Gy), may be closest to the optimal dose regimen currently available. Naturally, precise staging is an essential prerequisite for accurate treatment [82]. Among LS-SCLC patients, 15% will be upgraded to extensive-stage after PET-CT examination, while 5% of extensive-stage patients will be downgraded to limited-stage. ^18^F-FDG PET-CT examination should be advocated, which not only improves the accuracy of SCLC staging diagnosis but also plays an important role in guiding the precise delineation of the radiotherapy target [83].

## 9. Conclusions

HypoTRT and HyperTRT offer greater survival benefits compared to ConvTRT, mainly by shortening treatment duration. HyperTRT has certain advantages over HypoTRT in balancing acute toxic reactions, late toxic reactions, and treatment efficacy, but current research evidence does not show statistical significance. Based on the included research, the TRT regimen which is 60 Gy/40 F administered twice daily, provided more effective tumor control with acceptable toxic complications. Further prospective studies are required to continue investigating more suitable radiotherapy regimens.

## Figures and Tables

**Figure 1 cancers-16-01908-f001:**
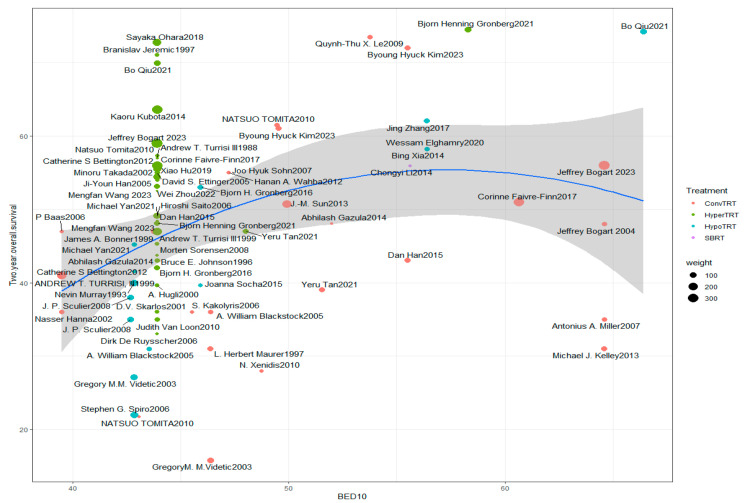
Scatter plot depicting the relationship between BED10 and 2-year overall survival rates for all treatment groups. The blue line and gray area represent the linear fitting results, serving as auxiliary lines to identify patterns. The fit *p*-value has no statistical significance [3,7,8,9,10,11,12,13,14,15,16,17,18,19,20,21,22,23,24,25,26,27,28,29,30,31,32,33,34,35,36,37,38,39,40,41,42,43,44,45,46,47,48,49,50,51,52,53,54,55,56,57,58,59].

**Figure 2 cancers-16-01908-f002:**
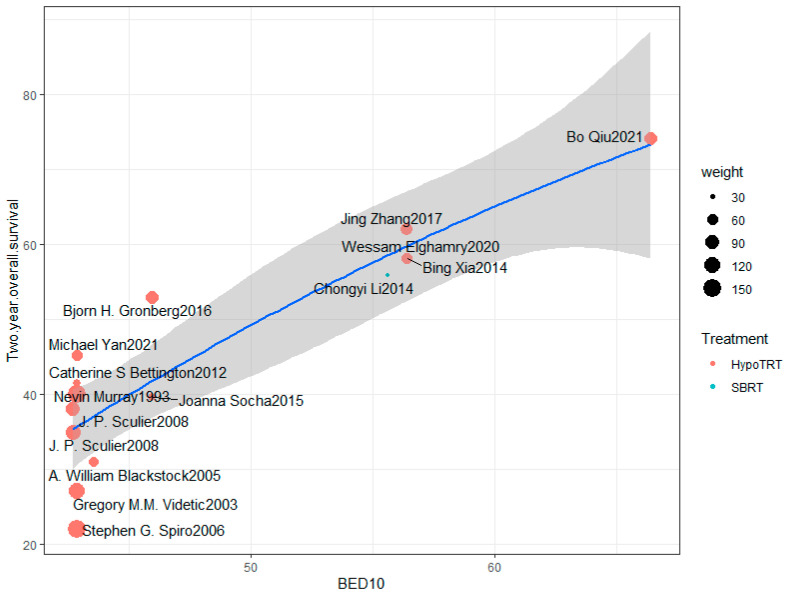
Scatter plot of 2-year overall survival versus BED10 in the HypoTRT treatment group. The blue line represents the quadratic equation fit, while the gray area depicts the confidence interval. R^2^ = 0.93, *p* < 0.0001 [14,16,42,44,48,49,52,53,54,55,56,57,58,59].

## Data Availability

The data in this study can be obtained from the published articles.

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
