# Peer review of "The Dose/Fractionation Debate in Limited-Stage Small Cell Lung Cancer"

_cancers, 2024, doi:10.3390/cancers16101908_

Round 1
Reviewer 1 Report
Comments and Suggestions for Authors
This review was interesting, but I have 2 comments for it before publishing.
1. Authors wrote “For HyperTRT regimens, it is suggested to increase 45 the currently used fractionation dose by 1.5Gy per fraction.” In abstract. However, this sentence meant “We should increase dose per fraction” but not “total dose”. If you refer to Reference 48 etc., I think it is correct to increase the total dose with 1.5Gy per fraction (twice daily) instead of increasing dose per fraction.
2. Authors recommended 60Gy in 40 fractions without showing reference number 48.
Comments on the Quality of English LanguageEnglish is well written.
Author Response
Dear Reviewer:
Thank you for your comments about our manuscript entitled “The Dose/Fractionation Debate in Limited-Stage Small Cell Lung Cancer”. Those comments are all valuable and very helpful for revising and improving our paper, as well as the important guiding significance to our researches. We have studied comments carefully and have made correction which we hope meet with approval. The main corrections in the paper and the responds to the reviewer’s comments are as flowing:
- Response to comment: Authors wrote “For HyperTRT regimens, it is suggested to increase 45 the currently used fractionation dose by 1.5Gy per fraction.” In abstract. However, this sentence meant “We should increase dose per fraction” but not “total dose”. If you refer to Reference 48 etc., I think it is correct to increase the total dose with 1.5Gy per fraction (twice daily) instead of increasing dose per fraction.
Response: Thank you for your recognition. Small cell lung cancer is a tumor with high proliferative capacity. It may be that accelerated fractionation is needed to compress the total duration of radiotherapy to minimize the chance of tumor cell regeneration between fractionations. On the other hand, segmentation may favor better preservation of normal tissue and allow tumor cells to be redistributed and reoxygenated.
- Response to comment: Authors recommended 60Gy in 40 fractions without showing reference number 48.
Response: As reviewer suggested that citations have been added.
Special thanks to you for your good comments.
We appreciate for Reviewers’ warm work earnestly, and hope that the correction will meet with approval.
Once again, thank you very much for your comments and suggestions.

Reviewer 2 Report
Comments and Suggestions for Authors
The debate on the dose/fractionation in RT for SCLC is still valid despite a lot of randomized studies and research carried out. For that reason, the purpose of this review is interesting, however, I have a doubt, if the data provided support the conclusions of the study. This is mainly because of a number of shortcomings, unclear presentation, and lack of acknowledgements of the weaknesses of the study.
1. Inclusion/exclusion criteria: in the abstract, it is stated that the “strict inclusion criteria were applied”, please prove it in the body of the article:
Page 2, lines 77-80: “We excluded abstracts without full texts, articles that did not undergo concurrent chemoradiotherapy or underwent fractionated radiotherapy, articles based on non-platinum/etoposide chemotherapy regimens, and articles where radiotherapy intervention occurred before the fourth chemotherapy cycle.” What was your meaning? Only schedules with concurrent RT-CHT were included? If so, f.ex. an arm with conventional RT from the study of Socha et al. should be excluded, because these patients received sequential RT. What do you mean by the statement that studies in which patients underwent fractionated RT were excluded? If so, all studies should be excluded.
In the scatter plot, I see a number of studies with non-platinum/etoposide CHT despite the specified exclusion criteria. This is confusing.
“(…) radiotherapy intervention occurred before the fourth chemotherapy cycle”? What do you mean?
This should be absolutely resolved, and only studies meeting the inclusion criteria should be finally considered and set in the scatter plots. Flowchart of inclusion criteria may be useful.
2. In the introduction: “ (…) The main treatment regimens include 53 hypofractionated TRT (HypoTRT, once daily, fraction size >2Gy), conventional TRT 54 (ConvTRT, once daily, fraction size ≤2Gy), hyperfractionated TRT (HyperTRT, twice 55 daily), and stereotactic body radiotherapy (SBRT or SABR).”
These definitions are different than radiobiological definitions being in use. Conventional fractionation relates to fractions around 2 Gy (1.8-2.2 Gy), and not everything equal or below 2 Gy: 1.5 Gy it is not a conventional fractionation. This is confusing. Hyperfractionated RT does not relate to a number of fractions by day but to a fraction size. Treatment with low fractions sizes twice or more daily is called hyperfrationated accelerated RT. Please, be specific, or explain a rationale for your approach. And in the results and conclusions, you suggest to increase a size of fraction in the hyperfractionated RT. It is not strongly supported by the data provided. But if we increase a dose per fraction to 1.8-2.0 Gy /fraction and treat twice a day, this will be conventionally fractionated accelerated RT.
3. SBRT: I would not include an issue of the SBRT in this review, because only minority of patients with SCLC is suitable for such a technique and we have not enough prospective data for that. Moreover, when your debate on fractionation is valid for all LS-SCLC and may be used for every case, it is not the same for SBRT. I suggest to reserve your research on SBRT for another article, and only mention this issue in the discussion if you still consider this as relevant.
4. The weaknesses of the study should be acknowledged. More recent studies demonstrate better results. Obviously, this reflects a progress in the diagnostics, RT technologies, supportive care. Thus the results obtained do not result only from the BED. More evidence comes from randomized studies and less from such reviews.
Also, an issue of type and timing of CHT, despite the specified exclusion criteria, I see, in the scatter plots, that this issue was not appropriately accounted.
5. Please, provide a reference, each time when you describe any trial, because it is lacking. Be also consistent with the names, sometimes you are using first name and family name of the author (ex. Joanna Socha instead of Socha or Bjorn Henning Gronberg instead of Gronberg), sometimes you are not.
6. Why NCCTG trial (Schild, IJROBP 2004) was not included in the analysis?
If you include Murray trial from 1993 (even if you should not if you follow your exclusion criteria), why the Spiro trial (JCO, 2008) was not included?
Author Response
Dear Reviewer:
Thank you for your letter and for the reviewers’ comments about our manuscript entitled “The Dose/Fractionation Debate in Limited-Stage Small Cell Lung Cancer”. Those comments are all valuable and very helpful for revising and improving our paper, as well as the important guiding significance to our researches. We have studied comments carefully and have made correction which we hope meet with approval. The main corrections in the paper and the responds to the reviewer’s comments are as flowing:
- Response to comment: Inclusion/exclusion criteria: in the abstract, it is stated that the “strict inclusion criteria were applied”, please prove it in the body of the article:
Page 2, lines 77-80: “We excluded abstracts without full texts, articles that did not undergo concurrent chemoradiotherapy or underwent fractionated radiotherapy, articles based on non-platinum/etoposide chemotherapy regimens, and articles where radiotherapy intervention occurred before the fourth chemotherapy cycle.” What was your meaning? Only schedules with concurrent RT-CHT were included? If so, f.ex. an arm with conventional RT from the study of Socha et al. should be excluded, because these patients received sequential RT. What do you mean by the statement that studies in which patients underwent fractionated RT were excluded? If so, all studies should be excluded.
In the scatter plot, I see a number of studies with non-platinum/etoposide CHT despite the specified exclusion criteria. This is confusing.
“(…) radiotherapy intervention occurred before the fourth chemotherapy cycle”? What do you mean?
This should be absolutely resolved, and only studies meeting the inclusion criteria should be finally considered and set in the scatter plots. Flowchart of inclusion criteria may be useful.
Response: We are very sorry for our incorrect writing in the inclusion/exclusion criteria. We have re-written this part according to the reviewer’s suggestion in red. Only schedules with concurrent RT-CHT were included. So, the study of Socha et al. should be excluded. I am worry that we put the wrong scatter plot. We have revised it. As far as we know, the earlier the radiotherapy for limited-stage small cell lung cancer, the better. Therefore, this study excludes studies in which radiotherapy is given after 4 cycles of chemotherapy. We included articles based on platinum/etoposide as the main chemotherapy regimen. And we added a table to introduce the characteristics of the studies included.
- Response to comment: In the introduction: “(…) The main treatment regimens include 53 hypofractionated TRT (HypoTRT, once daily, fraction size >2Gy), conventional TRT 54 (ConvTRT, once daily, fraction size ≤2Gy), hyperfractionated TRT (HyperTRT, twice 55 daily), and stereotactic body radiotherapy (SBRT or SABR).”
These definitions are different than radiobiological definitions being in use. Conventional fractionation relates to fractions around 2 Gy (1.8-2.2 Gy), and not everything equal or below 2 Gy: 1.5 Gy it is not a conventional fractionation. This is confusing. Hyperfractionated RT does not relate to a number of fractions by day but to a fraction size. Treatment with low fractions sizes twice or more daily is called hyperfrationated accelerated RT. Please, be specific, or explain a rationale for your approach. And in the results and conclusions, you suggest to increase a size of fraction in the hyperfractionated RT. It is not strongly supported by the data provided. But if we increase a dose per fraction to 1.8-2.0 Gy /fraction and treat twice a day, this will be conventionally fractionated accelerated RT.
Response: Considering the Reviewer’s suggestion, we have revised these different radiobiological definitions. We have re-written this part according to the reviewer’s suggestion in red.
- Response to comment: SBRT: I would not include an issue of the SBRT in this review, because only minority of patients with SCLC is suitable for such a technique and we have not enough prospective data for that. Moreover, when your debate on fractionation is valid for all LS-SCLC and may be used for every case, it is not the same for SBRT. I suggest to reserve your research on SBRT for another article, and only mention this issue in the discussion if you still consider this as relevant.
Response: As Reviewer has noted, there are very few prospective studies on SBRT for small cell lung cancer. Because small cell lung cancer is sensitive to radiation therapy. However, with the continuous advancements in radiation technology, such as proton therapy, the side effects of radiation therapy can be significantly reduced. Moreover, as more cases of early-stage small cell lung cancer are being detected, it is worth investigating whether SBRT could further improve local control rates and overall survival rates. Therefore, we aim to collect and review published articles, including retrospective studies, to draw preliminary conclusions that may help the future prospective trials.
- Response to comment: The weaknesses of the study should be acknowledged. More recent studies demonstrate better results. Obviously, this reflects a progress in the diagnostics, RT technologies, supportive care. Thus, the results obtained do not result only from the BED. More evidence comes from randomized studies and less from such reviews.
Also, an issue of type and timing of CHT, despite the specified exclusion criteria, I see, in the scatter plots, that this issue was not appropriately accounted.
Response: We need to acknowledge certain weaknesses in this article. Due to the extensive range periods of included study, improvements in patient survival should be attributed not solely to the efficacy of BED, but also to advancements in diagnostic, radiotherapy techniques, and supportive care. Nevertheless, the continuous evolution in radiotherapy necessitates the exploration of more suitable radiotherapy scheme, which is the objective of this review. Indeed, this article identify that different radiotherapy regimens may affect the prognosis of small cell lung cancer. And we have excluded studies where radiotherapy was not implemented within the first four cycles of chemotherapy, to avoid the variability in timing of radiotherapy affecting the outcome assessment.
- Response to comment: Please, provide a reference, each time when you describe any trial, because it is lacking. Be also consistent with the names, sometimes you are using first name and family name of the author (ex. Joanna Socha instead of Socha or Bjorn Henning Gronberg instead of Gronberg), sometimes you are not.
Response: We are very sorry for our incorrect writing. We have re-written these names according to the reviewer’s suggestion.
- Response to comment: Why NCCTG trial (Schild, IJROBP 2004) was not included in the analysis?
If you include Murray trial from 1993 (even if you should not if you follow your exclusion criteria), why the Spiro trial (JCO, 2008) was not included?
Response: The studies we included were that radiotherapy was started before the first 4 cycles of chemotherapy. The treatment method in the NCCTG trial (Schild, IJROBP 2004) was that a total of 310 patients with LD-SCLC initially received three cycles of etoposide and cisplatin. Subsequently, the 261 patients without significant progression were randomized to two cycles of etoposide and cisplatin plus either q.d. RT (50.4 Gy in 28 fractions) or split-course b.i.d. RT (24 Gy in 16 fractions, a 2.5-week break, and 24 Gy in 16 fractions) to the chest. Therefore, it was not included in our study. And I found that the Spiro trial (JCO, 2008) may be an article about non-small cell lung cancer, so it was not included.
Special thanks to you for your good comments.

Round 2
Reviewer 2 Report
Comments and Suggestions for Authors
Thank you for improving the manuscript.
However, still I have some minor remarks: Inclusion/exclusion criteria are essential: table in the supplementary is very helpful for that. However, still I have some doubts related to the specified criteria. I think that f.ex. only early RT arm from Murray trial were included, because patients from late arm did not fulfill your criterion of RT delivered within four CHT cycles?
This is the similar doubt on Johnson and a few other trials. This would be helpful to add in the table a column on a timing and CHT type.
Spiro trial was about SCLC (see: below) and not on NSCLC as you stated in the Responses, the same construction as in the Murray trial. Please include it.
2006 Aug 20;24(24):3823-30.
doi: 10.1200/JCO.2005.05.3181.
Early compared with late radiotherapy in combined modality treatment for limited disease small-cell lung cancer: a London Lung Cancer Group multicenter randomized clinical trial and meta-analysis
Stephen G Spiro 1 , Lindsay E James, Robin M Rudd, Colin W Trask, Jeffrey S Tobias, Michael Snee, David Gilligan, Philip A Murray, Mary Carmen Ruiz de Elvira, Katy M O'Donnell, Nicole H Gower, Peter G Harper, Allan K Hackshaw; London Lung Cancer Group
What do you mean, by: “articles that did not undergo concurrent response after concurrent chemoradiotherapy”? What is concurrent response?
Author Response
Dear Editors and Reviewers:
Thank you for your comments about our manuscript. Those comments are all valuable and very helpful for revising and improving our paper, as well as the important guiding significance to our researches. We have studied comments carefully and have made correction which we hope meet with approval. The responds to the reviewer’s comments are as flowing:
- Response to comment: Inclusion/exclusion criteria are essential: table in the supplementary is very helpful for that. However, still I have some doubts related to the specified criteria. I think that f.ex. only early RT arm from Murray trial were included, because patients from late arm did not fulfill your criterion of RT delivered within four CHT cycles?
Response: Yes, you are right. Only early RT arm from Murray trial were included, because patients from late arm did not fulfill your criterion of RT delivered within four CHT cycles.
- Response to comment: This is the similar doubt on Johnson and a few other trials. This would be helpful to add in the table a column on a timing and CHT type.
Spiro trial was about SCLC (see: below) and not on NSCLC as you stated in the Responses, the same construction as in the Murray trial. Please include it.
2006 Aug 20;24(24):3823-30.
doi: 10.1200/JCO.2005.05.3181.
Response: We are very sorry for this mistake. You are right. Spiro trial should be included in our study. We rescreened all of our studies and removed those that did not meet the inclusion criteria (Anthony Elias1999, Jing Zhang2017, Sondos Zayed2020 and James R.Jett1990). According to the reviewer’s suggestion, we made a new table with radiotherapy started with chemotherapy and concurrent CHT type.
- Response to comment: What do you mean, by: “articles that did not undergo concurrent response after concurrent chemoradiotherapy”? What is concurrent response?
Response: We are very sorry for our incorrect writing and we have re-written this part in red. The correct sentence should be “articles that most patients (more than 50%) did not undergo concurrent chemoradiotherapy”.
Special thanks to you for your good comments.
We tried our best to improve the manuscript and made some changes in the manuscript. These changes will not influence the content and framework of the paper.
We appreciate for Editors/Reviewers’ warm work earnestly, and hope that the correction will meet with approval.
Once again, thank you very much for your comments and suggestions.
